# Highly Efficient Copper Doping LaFeO₃ Perovskite for Bisphenol A Removal by Activating Peroxymonosulfate

**Xin Zhong** [1,2,*], **Junjie Liu** [3,*], **Haonan Jie** [2], **Wenting Wu** [2] and **Fubin Jiang** [1]

1 Experiment and Practice Innovation Education Center, Beijing Normal University at Zhuhai, Zhuhai 519088, China
2 Department of Environmental Engineering and Science, Beijing Normal University, Zhuhai 519088, China
3 Zhuhai Xijiang Water Construction Management Co., Ltd., Zhuhai Water Environment Holdings Group Ltd., Zhuhai 519008, China
* Correspondence: zhongxin@bnu.edu.cn (X.Z.); liujunjie213@sina.com (J.L.); Tel.: +86-(0)756-3621560 (X.Z.)

**Abstract:** A series of copper doping LaFeO₃ perovskite (LaCu$_x$Fe$_{1-x}$O₃, LCFO, x = 0.1, 0.4, 0.5, 0.6, 0.9) are successfully synthesized by the sol-gel method under mild conditions. In this study, it is applied for the activation of peroxymonosulfate (PMS) for bisphenol A (BPA) removal. More than 92.6% of BPA was degraded within 30 min at 0.7 g/L of LCFO and 10.0 mM of PMS over a wide pH range with limited leaching of copper and iron ions. The physical–chemical properties of the catalysts were demonstrated by using X-ray diffraction (XRD), N₂ adsorption–desorption isotherms, scanning electron microscopy (SEM), and X-ray photoelectron spectroscopy (XPS). Furthermore, the effects of catalyst dosage, PMS concentration, initial pH value, and inorganic anions on the LCFO/PMS system were fully investigated. Quenching experiments were performed to verify the formation of reactive oxidant species, which showed that the radical reaction and mechanisms play a great role in the catalytic degradation of BPA. The perovskite LCFO is considered a stable, easy to synthesize, and efficient catalyst for the activation of PMS for wastewater treatment.

**Keywords:** bisphenol A; LaCu$_x$Fe$_{1-x}$O₃ perovskite; peroxymonosulfate





## 1. Introduction

In recent years, pharmaceuticals and personal care products (PPCPs) have been widely used and could be easily found and detected in various environmental compartments at trace concentrations [1–6]. Bisphenol A (BPA), which is used as a common chemical reagent in the production of plastics and epoxy resins, can induce endocrine dyscrasia and cancer [7–9]. It poses great threat to human health even at low concentrations, due to which it has attracted increasing attention in the environmental protection areas. However, the BPA removal efficiency by conventional treatment technologies is unsatisfactory due to its trace concentration and negligible removal by only a single removal technology [10]. It is important to obtain environmental-friendly treatment technology for the degradation of BPA.

On the other hand, sulfate radical-based advanced oxidation processes (SR-AOPs) have gone through a long developing process for the removal of recalcitrant organic contaminants in water [11–14]. In SR-AOPs systems, many researchers have proven that organic contaminants are mineralized by reactive oxygen species (ROS), such as sulfate radical ($SO_4^{\bullet-}$), hydroxyl radical ($^{\bullet}OH$), and superoxide radical ($O_2^{\bullet-}$) [15–17]. The sulfate radical possessed a higher standard oxidative potential (2.6–3.1 V vs. NHE) than hydroxyl radical (1.8–2.7 V vs. NHE), a longer half-life, and a wider pH range in the wastewater treatment. Moreover, sulfate radicals could be more efficient in the removal of many refractory contaminants, which also result in higher selectivity in these processes than hydroxyl radicals [18]. In general, both peroxydisulfate (PS) and peroxymonosulfate (PMS) can be activated by multiple catalysts to form sulfate radicals. However, due to its asymmetric structure, PMS has shown better decomposing efficiency than PS. Various transition metals,

such as Fe [19–21], Cu [22–24], Mn [25–27], and Co [28–31], have been employed for the activation of PMS to generate ROS. Among the various catalysts, perovskites have been explored for use in dissociating PMS to remove organic contaminants from water to obtain high catalytic activity.

Perovskites have high thermal stability due to their $ABO_3$ formula, which gives them a unique oxidation state of the lattice oxygen in the perovskite materials. The A site cations consist of alkaline or rare earth metals ions where the B site can be consisted of transition metal ions [32–36]. Compared with the metal oxides catalysts, the structure and physicochemical properties endowed perovskites with flexible tuned structure to perform ideal catalyst functions for PMS activation with good catalytic activity. Perovskites can be easily separated from water to avoid secondary pollution to the environment, which can also prevent the aggregation problem in the metal oxides, showing good process stability in the process. Many studies have shown that partial substitution of B sites by transition metals can generate a defective perovskite structure, which is beneficial for catalytic activity related to the improvement of oxygen mobility [37].

It is known that cobalt ions are the most effective activator for PMS [38]. However, the Fe-based perovskites have been paid more attention due to low cost and low toxicity in contrast to Co-based perovskites. On the other hand, Cu-based perovskites can also efficiently decompose PMS in order to generate reactive oxidant species. It is really interesting to introduce copper ions in the $LaFeO_3$ perovskites for the degradation of pollutants [39]. The excellent catalytic properties of Cu-based and Fe-based perovskites, such as $La_2CuO_4$ [40], $LaFeO_3$ [41], and $LaFe_{1-x}Co_xO_3$ [42], can be attributed to the presence of foreign element maintained in the structure and modified physicochemical properties. The presence of foreign elements in the perovskites and transition metal species create excellent catalytic activity. The change in perovskites structure facilitate the electron transfers from the transition metals to PMS, producing more ROS in the process. The formed metal-O bonds can boost the electron transfer between the redox pairs of Fe(III)/Fe(II) and Cu(II)/Cu(I), leading to the improvement of catalytic activity. The perovskite $LaA_xB_{1-x}O_3$ (A or B = Co, Fe, Cu) is expected to be an efficient catalytic material in SR-AOPs [43]. In perovskite, the cation at both A and B sites can be substituted by a foreign one, making it feasible to control the valance state of cations without detriment of original crystal structure [44]. The capability of electron transfer is influenced by the La site and Fe/Cu sites in the redox reactions, which would be enhanced by the presence of the A–O–M bond, further leading to better catalytic activity. In this case, the Cu-doped $LaFeO_3$ perovskite catalyst would give an interesting insight into the synergistic strategy between bimetallic active ions in perovskite structure and the heterogeneous catalyst activating PMS to remove the refractory chemicals in water. To the best of our knowledge, studies focused on Cu-doped $LaFeO_3$ perovskites for PMS activation for BPA removal are still rare.

In this study, Cu doping $LaFeO_3$ perovskite oxide, $LaCu_xFe_{1-x}O_3$ (LCFO), was fabricated via the sol-gel methods and used as a catalytic activator for PMS to remove BPA from water. The physicochemical properties and morphology of LCFO have been systematically characterized by many techniques. In addition, the effects of different experimental conditions on the BPA degradation efficiency and different water matrices, including pure water, lake water, and sea water, have been tested in order to find the utility in actual water treatment. Further experiments, such as reusability, kinetics, and reaction mechanism, have been carried out to evaluate the actual application potential of LCFO. This research offers creative ideas about the high catalytic activity of LCFO in the decomposition of PMS, which provides a new way to illustrate the mechanism of the LCFO/PMS process in actual wastewater.

## 2. Results

### 2.1. Physicochemical Characteristics of LCFO

For the LCFO catalysts, the actual content of metals followed the named content. The X-ray powder diffraction patterns of the $LaCu_xFe_{1-x}O_3$ perovskites are shown in

Figure 1a,c,d. It was observed that with Cu doping into the structure of LaFeO₃, the perovskite phase is maintained without the diffraction peaks of copper oxides as the x = 0.1–0.5. While for x = 0.6 and 0.9 in the diffractograms, i.e., the copper content increased in the LaFeO₃ perovskite, some impurity peaks assigned to La₂CuO₄ (PDF card: 80–0063) and CuOₓ could be formed, as has been previously reported [45]. Accordingly, the impurity peak intensity in the composites increased as the amount of copper content increased. However, the characteristic diffraction peak of LaFeO₃ at 32.3° weakened as the copper content increased in LCFO-5. As shown in Figure 1c, the results show that the diffraction peaks corresponding to 121 planes shifted slightly toward the higher 2θ angle with the increase in x ratio. The results could be ascribed to the replacement of Fe by Cu element which aroused the increment of unit cell parameters due to the different ionic radii of copper and iron (Cu: 0.73 Å, Fe: 0.63 Å) [46–48]. The results confirmed the existence of copper and iron in the perovskite lattice. To estimate its catalytic stability, the XRD pattern of used LCFO-4 catalyst was also explored. No remarkable changes appeared in the XRD patterns of the used catalyst. However, it is observed that the intensity of peaks around ~29.2° and ~35.4° declined. The difference between fresh and used catalyst might be attributed to leaching metal ions which caused the loss of reactive sites in the original perovskite. The results demonstrate that the main perovskite structure remained during the reaction.

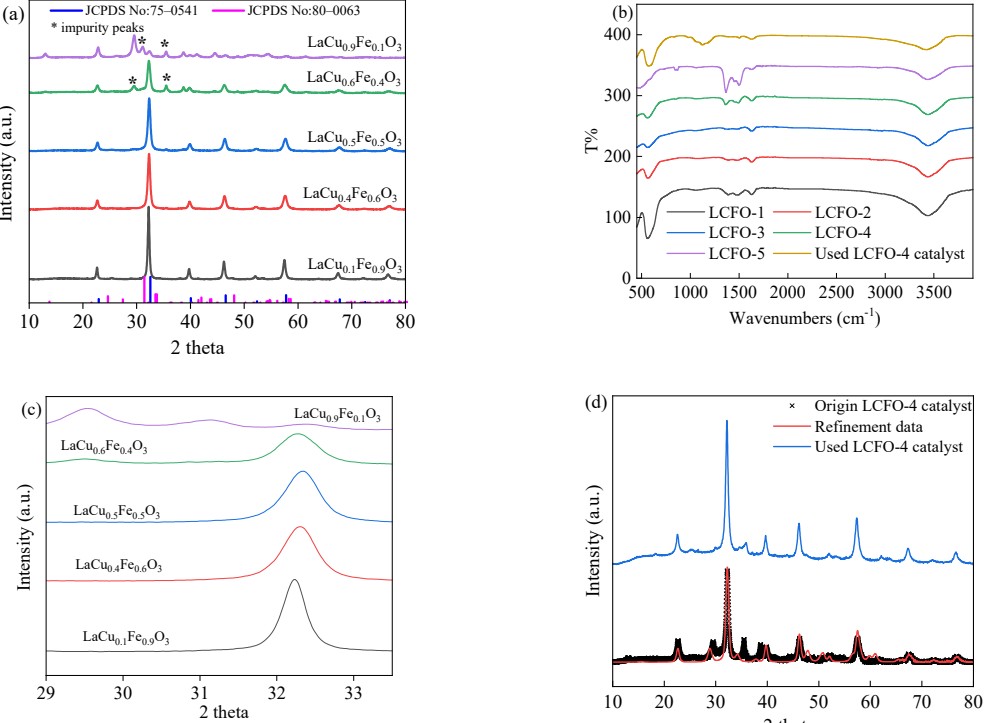

**Figure 1.** XRD patterns (**a**); and FTIR spectra (**b**) of LCFO catalyst, enlarged XRD patterns (**c**) and XRD patterns of refinement and used catalyst (**d**).

The FTIR spectra illustrate the surface functional groups in the LCFO catalyst and verified the doping Cu strategy in perovskite LaFeO₃. As shown in Figure 1b, the low intensity band at 563 cm$^{-1}$ and 439 cm$^{-1}$ is considered to represent the stretching vibrations of Fe–O and O–Fe–O deformation vibrations. In addition, the peak was observed slightly shifted toward higher wavenumbers with increasing Cu doping in the LaFeO₃ structure since the Fe was partially substituted by Cu [49–52]. For the used catalyst, the bands remained during the reaction with similar intensity.

The N₂ adsorption/desorption isotherms of LCO-4 catalysts are also summarized and shown in Figure 2a. The BET surface area of LCFO-4 was found to be 10.8 m²/g, and the

particle size was determined to be 40.4 nm. The type IV isotherm indicates the presence of mesopores in the catalyst, which could indicate good catalytic activity in the degradation of BPA.

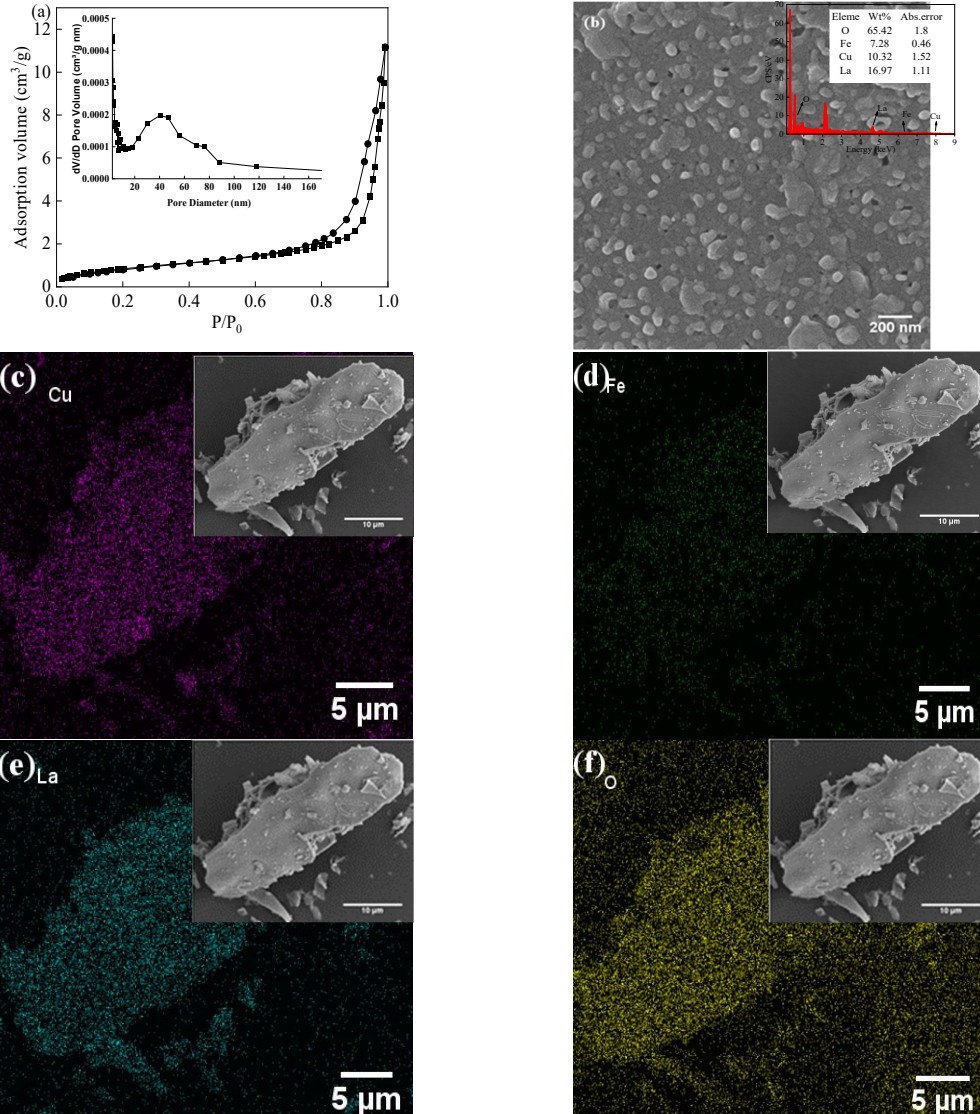

**Figure 2.** BET isotherm image (**a**) and SEM images and EDX spectra (**b**); Cu mapping (**c**); Fe mapping (**d**); La mapping (**e**); and O mapping (**f**) of LCFO-4.

The morphologies of the LCFO-4 catalyst were determined by SEM-EDS. As seen in Figure 2b–d, it was observed that the catalyst consists of a honeycomb-like structure with spheroidal particles in the size of 40–50 nm. An examination of the EDS spectra was also carried out to estimate the elemental composition of the nanoparticles. The La/Cu and La/Fe atomic ratios were calculated to be 0.6 and 0.4, respectively, which are very close to the stoichiometry atomic ratio of LCFO-4 (x = 0.6). Meanwhile, the atomic ratio O/Cu-Fe was determined to be 3.65, which is in good agreement with the stoichiometric ratio. The atomic mapping images also show that the four elements were highly dispersed over the as-prepared LCFO-4 sample [53].

The surface chemistry characteristics of the fresh LCFO-4 and used LCFO-4 were determined by examining the XPS spectra in Figure 3. In the XPS survey spectra, the characteristic peaks of four elements (La, Cu, Fe, O) were observed, which illustrate the chemical composition of the LCFO-4 sample. The XPS spectra of as-synthesized catalysts

showed the reaction roles of the Cu and Fe species that act as an electron acceptor [50–52]. It was observed that the peaks centered at 834.6 eV and 851.5 eV with corresponding satellite peaks located at 838.1 and 854.9 eV could be attributed to La 3d. For C 1s XPS spectra (Figure 3c), the peak centered at 284.6 eV could be assigned to the variable carbon adsorbed on the surface or the C–C coordination in the sample. And the peak at 288.7 eV was attributed to the bonded C=N or C–N of CN. The high-resolution Cu 2p and Fe 3p XPS spectra are shown in Figure 3d,e. The peaks and corresponding satellite peaks at approximately 709.8 eV and 723.3 eV were assigned to Fe species. Meanwhile, the peaks located at 932.4 eV indicate the presence of Cu(I) with the molar content of 58.7%, and the other peaks centered at 934.2 eV with the molar content of 41.3% belonged to Cu(II). The quantified atomic ratios of Cu(I) and Fe(II) are quantified as 41.8% and 32.1% in the used catalyst after use, respectively, indicating that the redox reaction between Cu and Fe occurred in the reaction. Four deconvoluted peaks at 528.5 eV, 529.4 eV, 531.6 eV, and 532.8 eV are found in the high-resolution of O spectra, which could be attributed to lattice oxygen, surface oxygen species, hydroxyl groups, and absorbed water, respectively.

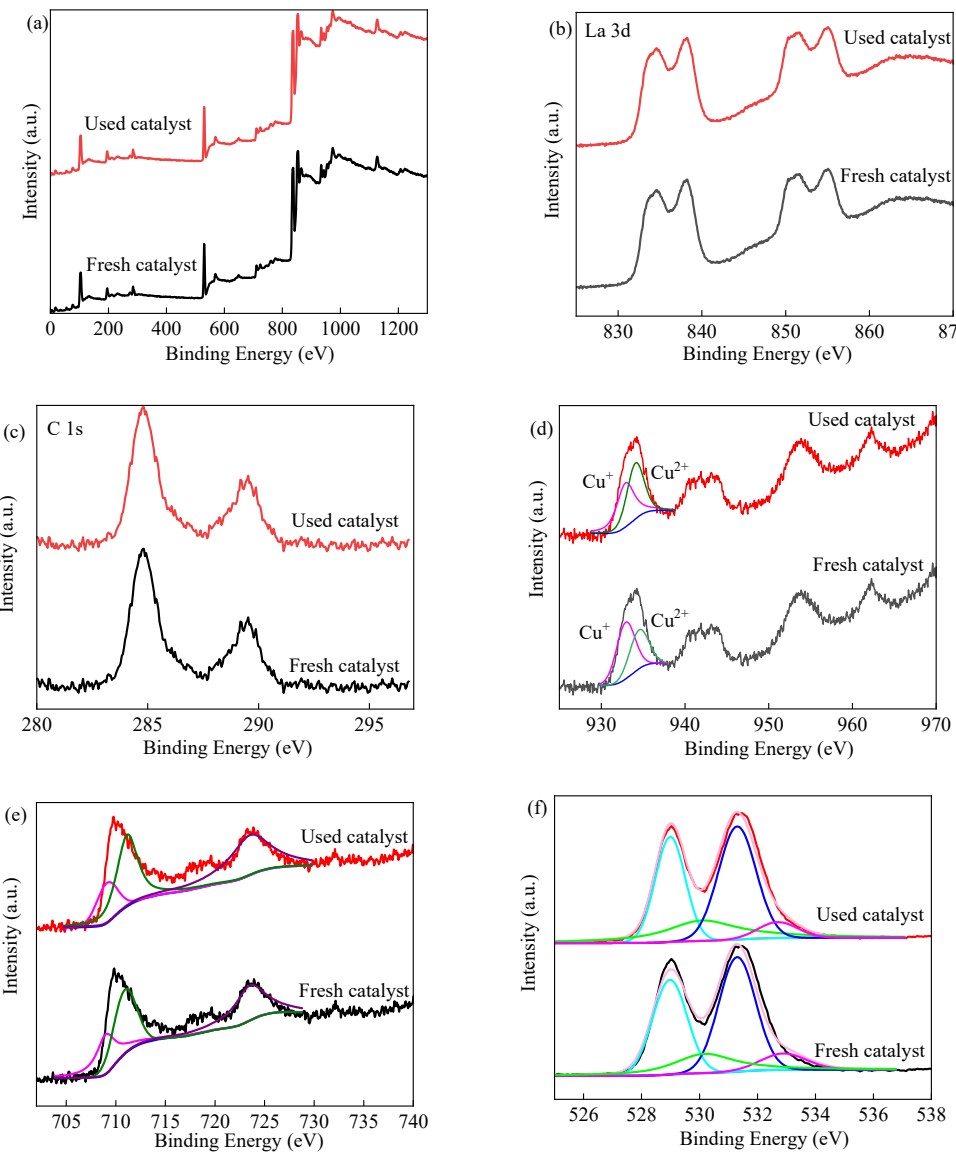

**Figure 3.** The deconvoluted XPS spectra of fresh and used LCFO-4: (**a**) survey spectra; (**b**) La 3d; (**c**) C 1s; (**d**) Cu 2p; (**e**) Fe 2p; (**f**) O 1s.

### 2.2. Catalytic Activity of Catalyst

To analyze the influence of the LCFO-4 activation for PMS on the degradation of BPA, the experiments were performed with the same experimental parameters previously used. As shown in Figure 4a, only a little removal of BPA (approximately 1.4%) was observed in the presence of LCFO-4 alone, which could be attributed to the low adsorption efficiency of BPA on the surface of LCFO-4. Moreover, the degradation efficiency was less than 8.3% in the 30 min reaction without the addition of the LCFO-4 catalyst by using PMS alone. The results indicated that BPA was hardly removed from the water by only PMS or only catalyst, indicating that the formation of reactive radicals played a great role in the SR-AOP processes [54–56]. With the increasing addition of copper element, the removal efficiency of BPA degradation performance of the samples increased first, but decreased with further addition of copper content. Compared to the other samples, LCFO-4 displayed the best performance. With the presence of pure $La_2CuO_4$ and pure $LaFeO_3$, the removal efficiency of BPA was 89.1% and 67.3% in 30 min reaction. The presence of PMS and LCFO-4 showed a great decline in BPA concentration (92.7%). The synergy between the impurity phases in the original LCFO-4 catalyst might affect the catalytic activity. Meanwhile, the activation of PMS by homogeneous copper and iron ions was responsible for 31.2% degradation of BPA. Therefore, it is reasonably speculated that due to the presence of LCFO-4, LCFO-4 was responsible for the activation of the PMS to generate reactive species for the high degradation efficiency of BPA.

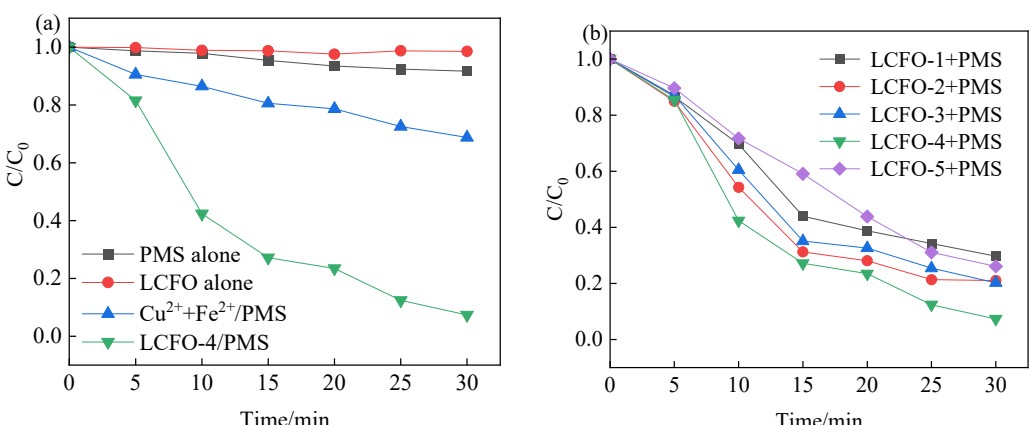

**Figure 4.** (**a**) The removal of BPA in different systems; (**b**) different Cu/Fe ratio contents. Conditions: $C_0$ = 0.05 mM, $[Cu^{2+}]$ = 0.859 mg/L, $[Fe^{3+}]$ = 0.048 mg/L, [LCFO] = 0.7 g/L, [PMS] = 10 mM, $pH_0$ 6.8.

### 2.3. Effect of Catalyst Dosage

It is studied that the heterogeneous catalyst played a key role in the formation of reactive species for PMS activation [57]. In order to investigate the effect of catalyst dosage on the activation of PMS, a series of catalyst dosage were employed in the heterogeneous catalyst/PMS process. Figure 5 show the effect of catalyst dosage on BPA removal efficiency and kinetic rate. When the dosage of LCFO-4 was varied from 0.1 g/L to 0.7 g/L, the BPA removal efficiency increased from 29.2% to 92.7% and corresponding k values increased from 0.0104 $min^{-1}$ to 0.0828 $min^{-1}$ within 30 min. While the catalyst dosage increased from 0.7 g/L to 1.0 g/L, although the remove efficiency reached to 94.3%, the increase in reaction rate constant increasement slowed down, as the corresponding k value increased from 0.0828 $min^{-1}$ to 0.0972 $min^{-1}$. This might be attributed to the fact that more active sites were provided in the reaction due to the increase in catalyst dosage, enhanced PMS activation, and the increased free radicals formed by activation of PMS. Therefore, the BPA removal efficiency and rate were both accelerated with the increase in catalyst dosage; when the amount of catalyst was greater than 0.7 g/L, more active sites were exposed in the reaction with more free radicals, where the quenching experiments would appear,

resulting in a reduced reaction rate. For the economic consideration, 0.7 g/L was chosen as the catalyst dosage for the sequenced experiments.

$$^{\bullet}OH + ^{\bullet}OH \rightarrow H_2O_2 \tag{1}$$

$$SO_4^{\bullet -} + SO_4^{\bullet -} \rightarrow S_2O_8^{2-} \tag{2}$$

$$SO_4^{\bullet -} + ^{\bullet}OH \rightarrow HSO_5^{-} \tag{3}$$

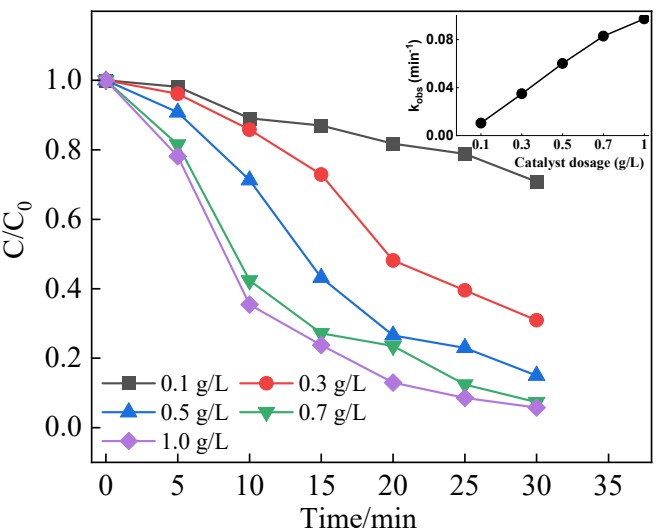

**Figure 5.** Effect of catalyst dosage on BPA removal. Conditions: $C_0$ = 0.05 mM, [PMS] = 10 mM, pH$_0$ 6.8.

### 2.4. Effect of PMS Concentration

On the other hand, a variation in PMS concentration would also affect the removal efficiency as the reaction oxidant [58]. As shown in Figure 6, with the PMS concentration ranging from 1.0 mM to 10 mM, the removal of BPA was significantly improved from 34.7% to 92.7%, and the reaction rate constant changed from 0.0138 min$^{-1}$ to 0.0828 min$^{-1}$. These results illustrated the fact that the LCFO-4 catalyst is an effective activator for PMS decomposition, leading to more reactive oxygen species during the reaction.

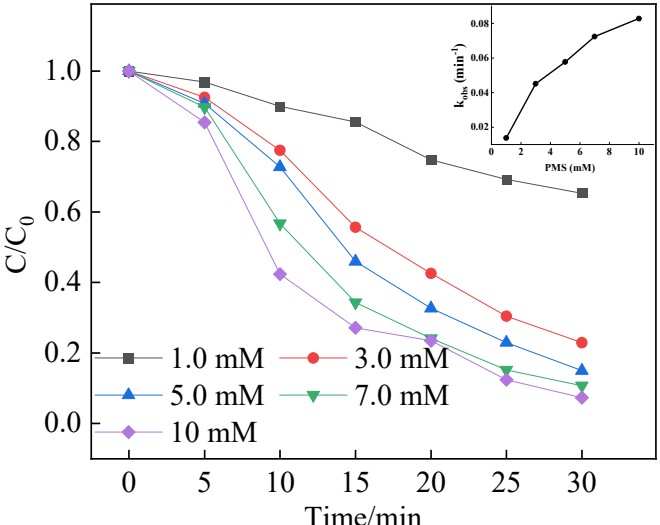

**Figure 6.** Effect of PMS concentration on BPA removal. Conditions: $C_0$ = 0.05 mM, [LCFO-4] = 0.7 g/L, pH$_0$ 6.8.

## 2.5. Effect of pH Value

The effect of the initial pH was investigated and shown in Figure 7. As the solution pH changed from 3.0 to 6.8, the BPA elimination process was enhanced, while the removal efficiency increased from 87.1% to 92.7%, and the reaction rate constant ranged from 0.0624 min$^{-1}$ to 0.0828 min$^{-1}$. It was studied that the solution pH values were quickly changed to around 4.5 during the whole reaction time under these pH values. This could be responsible for the similar BPA removal efficiency. The zeta potential was investigated and shown in Figure 7b. The isoelectric point of LCFO-4 is around 9.4, indicating the catalyst surface was charged when the pH is lower than 9.4. In this case, contact between $HSO_5^-$ and LCFO surfaces would be more feasible, which is the major PMS species under 9.4, favoring the following PMS activation. With the solution pH raised to 11.0, the degradation efficiency was further enhanced to the point where 100% removal efficiency was achieved in 30 min. These results show that the base activation conditions facilitated the decomposition of PMS, leading to the higher degradation efficiency, which is in good agreement with the previous studies [59–62]. On the other hand, the structure of the catalyst would be destroyed by the acidic conditions, leading to lower catalytic activity. The metal leaching was also tested under different pH values, as seen in Figure 7c. The results indicated that the LCFO-4 catalyst was stable in a wide range of pH, presenting good stability in the reaction.

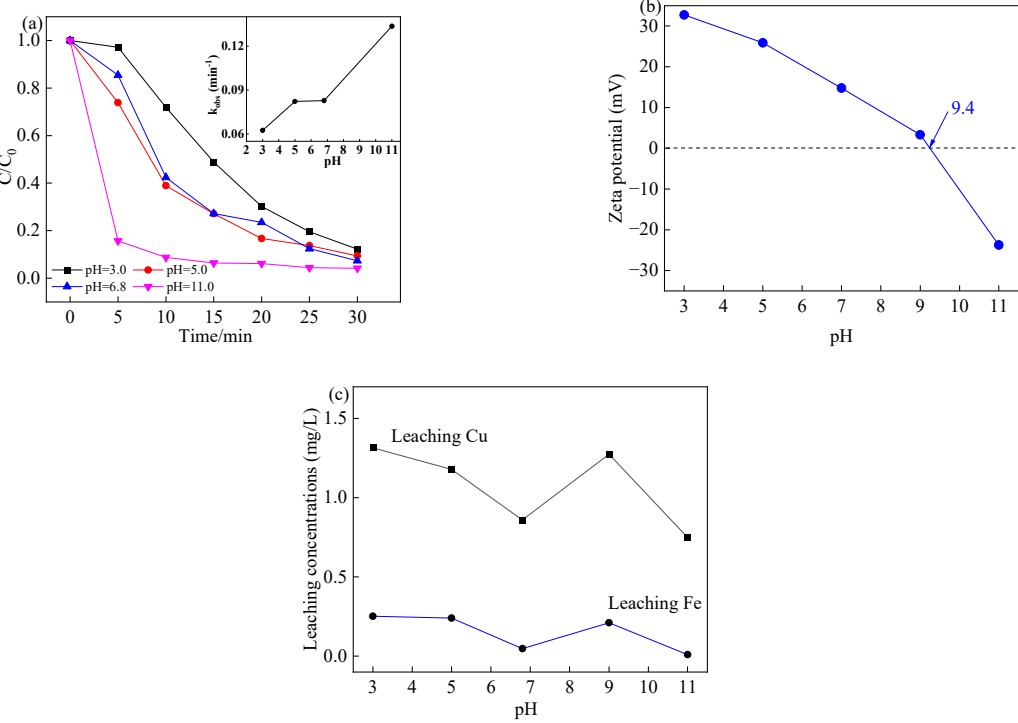

**Figure 7.** (**a**) Effect of pH value on BPA removal in LCFO-4/PMS systems; (**b**) zeta potential of LCFO-4; (**c**) metal leaching under different pH value. Conditions: $C_0$ = 0.05 mM, [LCFO-4] = 0.7 g/L, [PMS] = 10.0 mM.

## 2.6. Effect of Co-Existing Ions

Inorganic anions are considered as a reactant that can also compete with the reactive species or react with them to form new reactive species. In this study, the effects of Cl$^-$ and HCO$_3^-$ on BPA degradation were analyzed. As can be seen in Figure 8a, with the addition of chloride ions, the removal efficiency of BPA was approximately approached to 100% when the presence of 1.0 mM chloride ions was added in the systems. The reason could be ascribed to the reaction between Cl$^-$ and free radicals which caused the formation of $^\bullet$Cl and $^\bullet$ClOH, as shown in Equations (5)–(7) [63–66]. Despite the redox potential of these species is lower than that of hydroxyl radicals and sulfate radicals, the reaction

rate constants were high which showed good attack ability against BPA. In conclusion, lower concentration of $Cl^-$ showed good BPA removal efficiency in this system which is in agreement with the other reports [67–69]. On the other hand, with the presence of 5.0 mM and 10.0 mM $Cl^-$, it was observed that the BPA removal efficiency went through inhibition as the concentration of $Cl^-$ increased. The results could be ascribed to the side reaction between the $Cl^-$ and reactive species ($HSO_5^-$/$^\bullet OH$/$SO_4^{\bullet-}$) to generate chlorine species, which have lower redox potential values than hydroxyl radical and sulfate radical and lead to the consumption of the oxidant reactant.

$$Cl^- + HSO_5^- \rightarrow SO_4^{2-} + HClO \tag{4}$$

$$Cl^- + SO_4^{\bullet-} \rightarrow {}^\bullet Cl + SO_4^{2-} \tag{5}$$

$$Cl^- + {}^\bullet OH \rightarrow {}^\bullet ClOH \tag{6}$$

$${}^\bullet ClOH + H^+ \rightarrow {}^\bullet Cl + H_2O \tag{7}$$

$${}^\bullet Cl + Cl^- \rightarrow {}^\bullet Cl_2^- \tag{8}$$

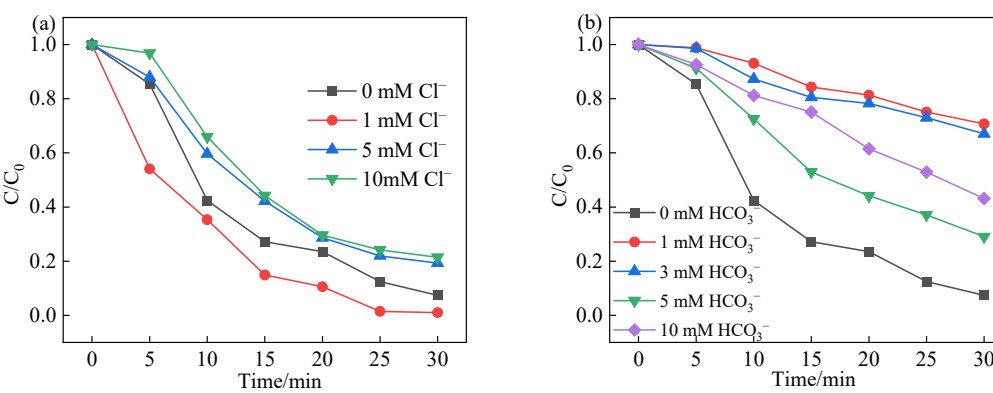

**Figure 8.** (**a**) Effect of $Cl^-$ on BPA removal; (**b**) effect of $HCO_3^-$ on BPA removal. Conditions: $C_0$ = 0.05 mM, [LCFO-4] = 0.7 g/L, [PMS] = 10.0 mM, $pH_0$ 6.8.

It has been reported that the presence of $HCO_3^-$ can capture the hydroxyl radicals and sulfate radicals, leading to the formation of less reactive carbonate radicals [64]. The degradation efficiency of BPA decreased from 91.0% to 42.0% within 30 min with increasing $HCO_3^-$ concentration from 0 to 5 mmol/L. The $k_{obs}$ values declined from 0.030 $min^{-1}$ to 0.0170 $min^{-1}$. The results could be ascribed to the fact that the solution pH value would change with the addition of $HCO_3^-$. When the addition of $HCO_3^-$ was 1.0 mM, 3.0 mM, 5.0 mM, and 10.0 mM, the solution pH was 3.25, 3.65, 4.31 and 5.97, respectively. Unlike the effect of choline ions, in the presence of $HCO_3^-$ anions, there is an obvious decrease in the catalytic activity of the system with the addition of $HCO_3^-$ compared to $Cl^-$. These results may be attributed to the rapid speed of bicarbonate anions with sulfate and hydroxyl radicals with the formation of lower redox potential ($HCO_3^{\bullet-}$/$CO_3^{\bullet-}$), leading to negative impact on BPA degradation. Based on the above reasons, $HCO_3^-$ presented stronger influence on the catalytic degradation than that of $Cl^-$.

$$^\bullet OH + HCO_3^- \rightarrow CO_3^{\bullet-} + H_2O \tag{9}$$

$$SO_4^{\bullet-} + HCO_3^- \rightarrow CO_3^{\bullet-} + SO_4^{2-} + H^+ \tag{10}$$

### 2.7. Identification of Reactive Oxidant Species

A series of quenching experiments were carried out to identify the dominant reactive radicals by using *tert*-butyl alcohol (TBA), *p*-benzoquinone (BQ), and methanol (MeOH) as scavenger of free radicals. It has been reported that TBA is an effective quenching chemical for $^\bullet OH$ (k = 4–9 × $10^5$ $M^{-1}s^{-1}$), whereas BQ is a scavenger for superoxide

(k = $2.9 \times 10^9$ M$^{-1}$s$^{-1}$), and MeOH is used for both $^\bullet$OH (k = $1.2$–$1.8 \times 10^9$ M$^{-1}$s$^{-1}$) and SO$_4^{\bullet-}$ (k = $1.6$–$7.8 \times 10^7$ M$^{-1}$s$^{-1}$) [65–67]. As can be seen in Figure 9, the results showed that the addition of TBA decreased the removal efficiency of BPA to 84.9%, and the addition of MeOH decreased the removal efficiency of BPA to 60.7%, implying both the sulfate radical and hydroxyl radical played a great role in the degradation of BPA in the PMS/LCFO system. Thus, the degradation efficiency decreased from 92.7% to 45.8% with the addition of BQ within 10 min, indicating that the superoxide was generated in the PMS/LCFO-4 system. These results can be ascribed to the scavenging effect of hydroxyl radicals to BQ and the superoxide suffered shorter half-life than that of sulfate radicals and hydroxyl radicals. In addition, the above results proved that both hydroxyl radicals and sulfate radicals are the important reactive oxidants in the heterogenous catalysis system.

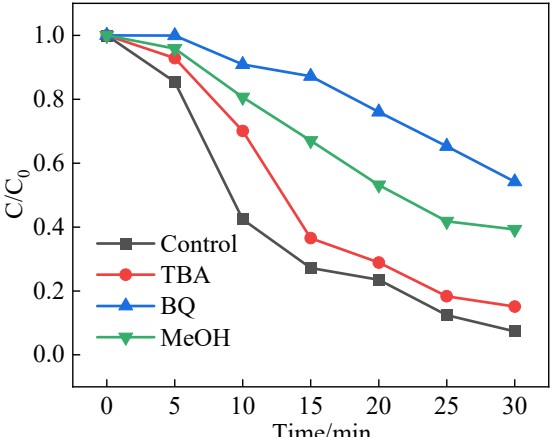

**Figure 9.** Effect of scavengers on BPA removal in the LCFO/PMS process. Conditions: C$_0$ = 0.05 mM, [LCFO-4] = 0.7 g/L, [PMS] = 10.0 mM, [Scavenger] = 50 mM.

## 3. Mechanism of the PMS Activation on LCFO

Reusability experiments were carried out to verify the stability of the LCFO catalyst. The degradation efficiencies were 92.5%, 87.6%, and 79.8% for three times, respectively, as shown in Figure 10a. Moreover, the k$_{obs}$ values are 0.0814 min$^{-1}$, 0.0723 min$^{-1}$, and 0.0605 min$^{-1}$, respectively, which are lower than that of fresh catalyst. Although the degradation efficiency was maintained in the recycle experiments, the rate constant was declined during the experiments. This could be ascribed to the leaching metal ions after each reaction since the loss of reactive sites. On the other hand, the reactive sites could be occupied by the produced intermediates which could also affect the degradation efficiency and the rate of speed of reaction.

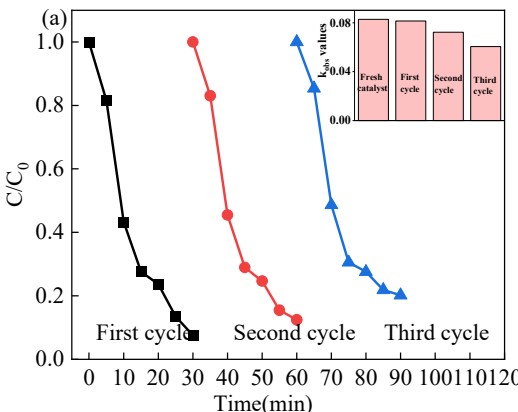 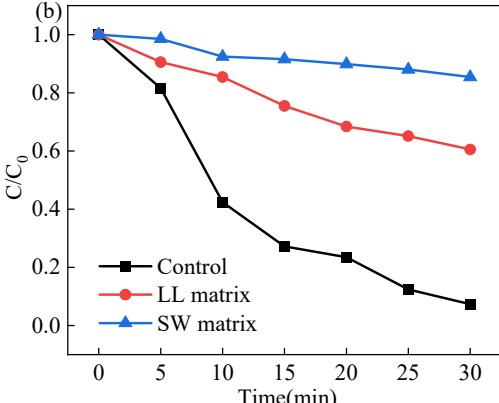

**Figure 10.** (**a**) Catalytic performance for three runs; (**b**) catalytic performance in different water matrix.

However, the typical degradation experiments were performed in pure water. In addition, the feasibility of using LCFO catalysts for practical applications was also carried out by investigating different water matrices, including Lake LiZe water (LL) and sea water (SW). As can be seen in Figure 10b, nearly 39.5% and 14.6% of BPA was removed from the LL and SW matrices, respectively. The removal efficiency was lower than that found in pure water, which is attributed to the coexisting ions, leading to the consumption of reactive species and oxidants. The LCFO-4/PMS system had showed high catalytic activity in different water matrices, demonstrating its feasibility and stability for use in various water matrices [68–70]. During the catalytic cycles, metal leaching was also explored to investigate the stability of LCFO catalyst. The leaching iron and copper ions were 0.048 and 0.859 mg/L after the catalytic reaction, which might be responsible for the reduced catalytic efficiency in the sequence usage. For investigating the contribution of homogeneous metal leaching, the experiment of leaching concentration of iron and copper ions was performed. An observed 31.2% degradation of BPA was obtained which is much lower than that in heterogeneous catalytic system. As a result, the contribution of homogeneous catalytic process was limited as the catalytic activation of PMS played a great role in the BPA removal.

TOC experiments were also used to investigate the mineralization of BPA under optimal reaction conditions. The TOC degradation was 56.8% in the LCFO-4/PMS system, showing good efficiency for the mineralization of BPA, which was in good agreement with the degradation efficiency of BPA. Moreover, the LCFO-4/PMS systems were also employed for the degradation of other organic contaminants, such as Orange II, Rhodamine B (RhB), and levofloxacin (LFX). The degradation efficiency values of Orange II and RhB were 100% in a 15 min reaction time, whereas the degradation efficiencies values of LFX were approximately 70.8%. The results showed that LCFO was an effective catalyst for many common organic pollutants, especially for the removal of dyes.

It has been reported that copper and iron elements are able to activate PMS to generate the sulfate radicals and hydroxyl radicals [69]. The La peak can be observed that there are two double peaks in the spectra. One peak was located at 834.6 eV with satellite peaks located at 838.1 eV, and the other peak was located at 851.5 eV with satellite peaks located at 854.9 eV. There is nearly no change in La valence of LCFO-4 before and after the catalytic reactions. The Cu 2p peaks can be divided into two peaks with binding energies of 932.4 eV and 934.2 eV, which indicate the presence of Cu(I) with the content of 58.7%. The peak observed at 709.8 eV with a satellite peak at 709.3 eV in the XPS spectra of Fe 2p was assigned to Fe 2p 1/2. The peak of Fe 2p 3/2 centered at 721.3 eV was considered to be Fe(III) species. In the used LCFO-4 catalysts, the relative Cu(I) and Fe(II) contents decreased after catalytic process, indicating the presence of electron transfer between the surface of the catalyst and the PMS. The O 1s XPS spectra of fresh and used LCFO-4 can be deconvoluted into four peaks located at 528.5, 529.4, 531.6, and 532.8 eV, which are responsible for the $O^{2-}$, absorbed oxygen species, and the hydroxyl groups of the adsorbed $H_2O$. The $O^{2-}$ peak content slightly decreased while the absorbed oxygen increased in the used catalyst, indicating that the $O^{2-}$ participated in the catalytic performance.

Based on these results and previous study, the mechanism of the LCFO-4 activation of PMS for BPA removal was inferred [70]. At first, PMS can be directly activated by Cu(II)/Cu(I) and Fe(III)/Fe(II) on the surface of LCFO catalyst, leading to the generation of sulfate radicals and hydroxyl radicals. In the meantime, the $HSO_5^-$ could be converted to hydroxyl sites through the hydrogen bonds which formed by the reaction between copper/iron states and $H_2O$ molecules. The $HSO_5^-$ was further reduced to sulfate radicals and combined to generate sulfate radicals. Moreover, the vacancies in the structure also played a great role in PMS activation, which led to the presence of redox reactions and facilitated the electron transfer with PMS on the surface of the LCFO catalyst. Moreover, non-radical mechanism was also detected in this process. Through the reaction between $HSO_5^-$ absorbed on LCFO and vacancies, the $O_o^\times$ was formed and participated in the reaction which represents the oxygen ions in a normal oxygen site. During the generation of $O_o^\times$, the singlet oxygen was formed. On the other hand, singlet oxygen ($^1O_2$) was also

generated by the self-decomposition of PMS, which is very slow and not detect in this study due to its low concentration. These reactive species noted above, can participate in the conversion removal of BPA to intermediates or end with full mineralization.

$$\text{Catalyst-Cu(I)} + \text{HSO}_5^- \rightarrow \text{Catalyst-Cu(II)} + \text{SO}_4^{\bullet-} + {}^\bullet\text{OH} \tag{11}$$

$$\text{Catalyst-Cu(II)} + \text{HSO}_5^- \rightarrow \text{Catalyst-Cu(I)} + \text{SO}_5^- + \text{H}^+ \tag{12}$$

$$\text{Catalyst-Fe(II)} + \text{HSO}_5^- \rightarrow \text{Catalyst-Fe(III)} + \text{SO}_4^{\bullet-} + {}^\bullet\text{OH} \tag{13}$$

$$\text{Catalyst-Fe(III)} + \text{HSO}_5^- + \text{H}^+ + \text{O}^* \rightarrow \text{Catalyst-Fe(II)} + \text{SO}_5^- + \text{H}_2\text{O} \tag{14}$$

$$\text{Catalyst-Fe(II)-OH}^- + \text{HSO}_5^- \rightarrow \text{Catalyst-Fe(III)-OH}^- + \text{SO}_4^{\bullet-} \tag{15}$$

$$\text{Catalyst-Cu(I)-OH}^- + \text{HSO}_5^- \rightarrow \text{Catalyst-Cu(II)-OH}^- + \text{SO}_4^{\bullet-} \tag{16}$$

$$\text{HSO}_5^- + {}^\bullet\text{OH} \rightarrow \text{SO}_5^{\bullet-} + \text{H}_2\text{O} \tag{17}$$

$$\text{SO}_5^{\bullet-} + \text{SO}_5^{\bullet-} \rightarrow \text{SO}_4^{\bullet-} + \text{SO}_4^{\bullet-} + \text{O}_2 \tag{18}$$

$$\text{HSO}_5^- + \text{O}^* \rightarrow \text{HSO}_4^- + {}^1\text{O}_2 \tag{19}$$

$$\text{HSO}_5^- + \text{SO}_5^{2-} \rightarrow \text{HSO}_4^- + \text{SO}_4^{2-} + {}^1\text{O}_2 \tag{20}$$

$$\text{SO}_4^{\bullet-} + \text{H}_2\text{O} \rightarrow {}^\bullet\text{OH} + \text{SO}_4^{2-} + \text{H}^+ \tag{21}$$

## 4. Materials and Methods

### 4.1. Chemicals

The chemical reagents were all analytical grade and used without further purification. The solutions were prepared with deionized water. Different water matrices, such as lake water (LiZe Lake in Zhuhai) and sea water (Zhuhai), were also introduced in the experiments for which only filtration of the floating impurities was done.

### 4.2. Preparation of LCFO

The sol-gel method was employed to synthesize LCFO particles. First, 10 mmol citric acid was dissolved in 5 mL deionized water. Then, 5 mmol La(NO$_3$)$_3\cdot$6H$_2$O was added into the solution with a certain amount of Cu(NO$_3$)$_2\cdot$3H$_2$O and Fe(NO$_3$)$_3\cdot$9H$_2$O. The homogenous solution was slowly heated to 80 °C to evaporate the water and continuously stirred for 2 h until the gel was found. The gel was dried at 110 °C in an oven, and calcined at 250 °C for 4 h to obtain a brown powder. The powder was secondarily calcined at 700 °C for 4 h. The serious LCFO catalyst was denoted as LCFO-1 (x = 0.1), LCFO-2 (x = 0.4), LCFO-3 (x = 0.5), LCFO-4 (x = 0.6), LCFO-5 (x = 0.9).

### 4.3. Experimental Procedures

In this study, a certain amount of PMS was added to a 250 mL glass beaker containing 100 mL of a 0.05 mM BPA solution at a neutral pH value. The pH of the solution was adjusted by 0.1 M H$_2$SO$_4$ and 0.1 M NaOH. The reaction was performed at room temperature (25 ± 2 °C) with magnetic stirring at 600 rpm. First, 0.5 g/L LCFO catalyst was added to the BPA solution under stirring. At time intervals of 5 min, 1 mL samples were collected by syringe and filtered through 0.22 μm membranes to remove the catalyst, and then the sample was mixed with 10 μL methanol (MeOH) to terminate the reactive radical reaction before analysis by high-performance liquid chromatography (HPLC, LC16, Shimadzu, Kyoto, Japan). The experiments were performed three times, and the average value was used.

### 4.4. Catalyst Characterization and Analytical Methods

The concentration of BPA was measured via a high-pressure liquid chromatography (HPLC) system (LC16, Shimadzu, Kyoto, Japan) equipped with a C18 (4.6 mm × 150 mm, 5 mm) column and a UV detector set at 280 nm. The mobile phase was 70% methanol and

30% ultrapure water, and the flow was set to 1.0 mL/min. To verify the reaction kinetics of BPA degradation in the LCFO/PMS process, a pseudo first-order kinetic model was employed. The degradation reaction rate constant of BPA was determined according to Equation (1):

$$\ln(C_0/C_t) = k_{obs} \times t \tag{22}$$

where $C_0$ is the initial BPA concentration of 0.05 M, $C_t$ is the BPA concentration during the degradation process, $k_{obs}$ represents the pseudo first-order reaction rate constant ($\min^{-1}$), and t is the reaction time.

X-ray diffraction (XRD) patterns were obtained using a Bruker D8 ADVANCE X-ray diffractometer (Bruker AXS, Karlsruhe, Germany) with graphite monochromatic Cu K$\alpha$ radiation ($\lambda$ = 1.54 Å) at an accelerating voltage of 40 kV and a current of 30 mA over a $2\theta$ scanning range of 10–80°. Scanning electron microscopy (SEM) was performed on a Hitachi SU8220 (Hitachi Corporation, Tokyo, Japan) field emission X-ray energy dispersive spectra (EDS) instrument to determine the element distribution. X-ray photoelectron spectroscopy (XPS) was conducted to obtain surface chemical information with a Thermo Fisher ESCALAB250Xi (Thermo Fisher Scientific, Waltham, MA, USA). The diffuse reflectance spectra (DRS) were obtained using an ultraviolet-visible-near infrared spectrophotometer with Shimadzu UV-3600 Plus (Shimadzu, Kyoto, Japan). The textural properties of the samples were analyzed by $N_2$ adsorption–desorption technology using an automatic Micromeritics Instrument Corporation TriStar II 3020 (Micromeritics Instrument Corporation, Norcross, GA, USA). The TOC analyzer (XPERT-TOC/TNb, Trace Elemental Instruments, Delft, The Netherlands) was introduced to determine the mineralization level of BPA. In the typical quenching experiments, methanol (MeOH) is employed to capture hydroxyl radicals ($^{\bullet}$OH) and sulfate radicals ($SO_4^{\bullet-}$). *Tert*-butyl alcohol (TBA) is used for scavenging hydroxyl radicals ($^{\bullet}$OH), and benzoquinone (BQ) is used to scavenge superoxide radical ($O_2^{\bullet-}$).

## 5. Conclusions

In this study, the perovskite $LaCu_xFe_{1-x}O_3$ was successfully synthesized through the sol-gel method and showed excellent catalytic activity for the removal of BPA. The BPA removal efficiency was approximately 92.6% in 30 min with the addition of 10.0 mM PMS and 0.7 g/L CFO-4. The hydroxyl radical ($^{\bullet}$OH) and sulfate radicals ($SO_4^{\bullet-}$) are considered as the main reactive species in this system. The stability and reusability of the catalytic performance were verified through three reaction cycles with limited amounts of copper and iron ions leaching into the solution. In summary, this study provides an efficient strategy to determine the catalytic potential of partial substitution of B site in perovskite for PMS activation toward the removal of refractory organic chemicals in wastewater treatment.

**Author Contributions:** Methodology, X.Z.; investigation, W.W., H.J. and X.Z.; resources, X.Z. and F.J.; writing—original draft preparation, X.Z.; writing—review and editing, X.Z.; project administration, X.Z., J.L. and F.J. All authors have read and agreed to the published version of the manuscript.

**Funding:** This work is financially supported by the Project of Innovative Foundation of Guangdong Province of China (Grant No. 2022KTSCX206), Project of Innovative Foundation of Guangdong Province of China (Grant No. 2020KTSCX177), Promotion Project Funds of Beijing Normal University, Zhuhai (Grant No. 201850001), Quality Engineering Project of Beijing Normal University, Zhuhai (Grant No. 201832) and Plan of Youth Teachers of Guangdong Higher Education Association (Grant No. 19GYB060).

**Data Availability Statement:** The authors confirm that the data supporting the findings of this study are available within the article.

**Acknowledgments:** We acknowledge the financial support from the Project of Youth Innovative Foundation of Guangdong Province of China, Science Promotion Project Funds of Beijing Normal University, Zhuhai, Quality Engineering Project of Beijing Normal University, Zhuhai and Plan of Youth Teachers of Guangdong Higher Education Association for help in XRD, SEM, TEM, and XPS analysis.

**Conflicts of Interest:** The authors declare no conflict of interest.

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
