# Peer review of "Highly Efficient Copper Doping LaFeO3 Perovskite for Bisphenol A Removal by Activating Peroxymonosulfate"

_catalysts, doi:10.3390/catal13030575_

Round 1
Reviewer 1 Report (Previous Reviewer 2)
accept
Author Response
We really appreciate your suggestions on our manuscript to help us improve the quality. We cherish your approval on our revised manuscript. We appreciate for reviewer’s warm advice earnestly. Once again, thank you very much for your comments and suggestions.Reviewer 2 Report (Previous Reviewer 3)
I am impressed by the significant improvements by the authors. Though, I still have some comments to be addressed before acceptance. Especially the purity of the synthesized perovskite (point 2 below) should be addressed.
1. Line 78-80 is a repetition of line 59-61.
2. In Fig. 1d, the difference in diffractograms are remarkable in contrast to the statement by the authors in line 107-110. First of all, in the original LCFO-4, 3-4 peaks are not assigned based on the Refinement. Second, the peaks at ~29 and ~35 decreases significantly in intensity. Please comment on this. Especially as it seems from the original perovskite that it is not a single-phase perovskite and a two(or more)-phase system might lead to better catalytic activity due to synergies between the phases. It is also previously published that 40% Cu-doping of LaFeO3 cannot be synthesized in single-phase, see: doi:10.1016/j.matchemphys.2009.05.020
3. Is the LaFe1-xCuxO3 composition in combination with PMS promising for the future? The degradation rate is lower than for another Fe-based perovskite without PMS. doi:10.3390/catal12030265
4. I suggest to add the kinetic konstant for comparison in the cycle-experiment in Fig. 9a. This gives - in my opinion - a better comparison than the %-degradation in 30 min.
Author Response
Please find the following response to the comments of referees. Response to the reviewer’s comments. (Reviewer's comments marked in blue) Reviewers' comments: Reviewer #2: I am impressed by the significant improvements by the authors. Though, I still have some comments to be addressed before acceptance. Especially the purity of the synthesized perovskite (point 2 below) should be addressed. Response: Thank you for your kind suggestions and encourage. We really appreciate your suggestions on our manuscript to help us improve the quality. We cherish your approval on our revised manuscript. We appreciate for reviewer’s warm advice earnestly. Once again, thank you very much for your comments and suggestions. 1 Line 78-80 is a repetition of line 59-61. Response: Thank you for your kind suggestions. We are sorry for the repetition of texts in line 78-80. The repetition texts are deleted and the reference was correct to the right place. The whole revised manuscript was check again to avoid the similar mistakes. Once again, thank you very much for your comments and suggestions. 2 In Fig. 1d, the difference in diffractograms are remarkable in contrast to the statement by the authors in line 107-110. First of all, in the original LCFO-4, 3-4 peaks are not assigned based on the Refinement. Second, the peaks at ~29 and ~35 decreases significantly in intensity. Please comment on this. Especially as it seems from the original perovskite that it is not a single-phase perovskite and a two(or more)-phase system might lead to better catalytic activity due to synergies between the phases. It is also previously published that 40% Cu-doping of LaFeO3 cannot be synthesized in single-phase, see: doi:10.1016/j.matchemphys.2009.05.020. Response: Thank you for your kind suggestions. As reviewer suggested that the refinement data by Jade 6.5 program have been carefully checked and modified. The renewed Fig.1d is presented in the revised manuscript. The difference between fresh and used catalyst might attributed to leaching metal ions which caused the loss of reactive sites and or the impurity phase in the original perovskite. The main perovskite maintained during the reaction. In the original perovskite, we used Jade 6.5 program to do the phase analysis and find the impurity peaks in the raw data. We synthesized the pure La2CuO4 and LaFeO3 to find the catalytic efficiency for PMS activation in BPA removal. With the presence of pure La2CuO4 and LaFeO3, the removal efficiency was 89.1% and 67.3% in 30 min reaction. The degradation efficiency was 92.7% with the presence of LCFO-4 and PMS. The synergy between the impurity phases in the original catalyst might affect the catalytic activity. The relevant reference was cited as Ref. [44]. The revised manuscript was listed below. ‘The X-ray powder diffraction patterns of the LaCuxFe1-xO3 perovskites are shown in Fig. 1(a, c, d). It was observed that with the Cu doping into the structure of LaFeO3, the perovskite phase is maintained without the diffraction peaks of copper oxides as the x=0.1-0.5. While for x=0.6 and 0.9 in the diffractograms, some impurity peaks as-signed to La2CuO4 (PDF card: 80-0063) could be found which is in agreement with previous study [44]. As shown in Fig.1c, the results show that the diffraction peaks corresponding to 121 planes shifted slightly toward higher 2θ angle as the increase of x ratio. The results could be ascribed to the replacement of Fe by Cu element which aroused the increment of unit cell parameters since the different ionic radius of copper and iron (Cu: 0.73 Å, Fe: 0.63 Å) [45, 46]. The results confirmed the existence of copper and iron in the perovskite lattice. To estimate its catalytic stability, the XRD pattern of used LCFO-4 catalyst was also explored. No remarkable changes appeared in the XRD patterns of the used catalyst. However, it is observed that the intensity of peaks around ~29.2o and ~35.4o was declined. The difference between fresh and used catalyst might attributed to leaching metal ions which caused the loss of reactive site and or the impurity phase in the original perovskite. The results demonstrate that the main perovskite structure remained during the reaction.’ Special thanks to you for your good comments to help us improve the manuscript. 3 Is the LaFe1-xCuxO3 composition in combination with PMS promising for the future? The degradation rate is lower than for another Fe-based perovskite without PMS. doi:10.3390/catal12030265 Response: Thank you for your kind suggestions. As reviewer suggested that the degradation rate constant might lower than that of in other Fe-based perovskite without the presence of PMS. However, it is still interesting to find out the synergistic effect between bimetallic active sites in perovskite B site of La-based perovskite structure. To the best of our knowledge, the rate constant of copper doping LaFeO3 was similar in the similar perovskite materials for PMS activation [1-3], despite the target pollutants were not the same. The word ‘promising future’ was not suitable in this paper and deleted in the revised manuscript. The reference was cited in the introduction part as [36]. Special thanks to you for your good comments to help us improve the manuscript. [1] G. Wang, C. Cheng, J. Zhu, L. Wang, S. Gao, X. Xia. Enhanced degradation of atrazine by nanoscale LaFe1-xCuxO3-δ perovskite activated peroxymonosulfate: Performance and mechanism. Science of the Total Environment, 2019, 673, 565-575. [2] Y. Rao, Y. Zhang, J. Fan, G. Wei, D. Wang, F. Han, Y. Huang, Jean-Philippe Croue. Enhanced peroxymonosulfate activation by Cu-doped LaFeO3 with rich oxygen vacancies: Compound-specific mechanisms. Chemical Engineering Journal, 2022, 432, 134882. [3] Y. Zhao, B. Huang, H. An, G. Dong, J. Feng, T. Wei, Y. Ren, J. Ma. Enhanced activation of peroxymonosulfate by Sr-doped LaFeO3 perovskite for Orange I degradation in the water. Separation and Purification Technology, 2021, 256, 117838. 4. I suggest to add the kinetic konstant for comparison in the cycle-experiment in Fig. 9a. This gives - in my opinion - a better comparison than the %-degradation in 30 min. Response: Thank you for your kind suggestions. We have made modifications according to the reviewer’s comments. Considering the reviewer’s warm advice, we have rewritten the part with the addition of kinetic constant for each recycle experiments. And the Fig.9a is also updated in the revised manuscript. The revised texts are below: ‘Moreover, the kobs values was 0.0814 min-1, 0.0723 min-1 and 0.0605 min-1, respectively, which are lower than that of fresh catalyst. Although the degradation efficiency was maintained in the recycle experiments, the rate constant was declined during the experiments. This could be ascribed to the leaching metal ions after each reaction since the loss of reactive sites. On the other hand, the reactive sites could be occupied by the produced intermediates which could also affect the degradation efficiency and the rate speed of reaction.’ We have studied comments carefully and have made correction which we hope meet with approval. We also checked the whole manuscript to avoid the incorrected writing, and marked red in the revised manuscript. Thank you again for your kind advice.
Reviewer 3 Report (Previous Reviewer 4)
I think the authors improved their paper sufficiently by performing additional experiments and analyzing the data.
Author Response
We really appreciate your suggestions on our manuscript to help us improve the quality. We cherish your approval on our revised manuscript. We appreciate for reviewer’s warm advice earnestly. Once again, thank you very much for your comments and suggestions.Round 2
Reviewer 2 Report (Previous Reviewer 3)
The authors responded to all comments. I still believe the XRD should be explained in more detail as the peak at ~35 exist in the LCFO-4 sample, but NOT in the refinement, thus , this must be an impurity.
Author Response
Reviewer #2: The authors responded to all comments. I still believe the XRD should be explained in more detail as the peak at ~35 exist in the LCFO-4 sample, but NOT in the refinement, thus, this must be an impurity. Response: Thank you for your kind suggestions and encourage. As reviewer suggested that the XRD data for LCFO-4 was carefully checked and modified in the revised manuscript. The peak around 29.4o and 35.6o in the LCFO-4 catalyst would be recognized as the impurity peaks of La2CuO4 and CuOx in the perovskite structure where the copper content was higher than x=0.6. The excess copper in the precursor led to the formation of La2CuO4 and CuOx. The intensity of the diffraction peaks of impurity species gradually strengthened in LCFO-5; however, the characteristic diffraction peak of LaFeO3 at 2θ = 32.3° weakened as the copper dose increased. The Fig.1a has been updated by adding the impurity signals which is marked yellow in the revised manuscript. The revised manuscript was marked red in the manuscript and listed below. ‘While for x=0.6 and 0.9 in the diffractograms, i.e., the copper content increased in the LaFeO3 perovskite, some impurity peaks assigned to La2CuO4 (PDF card: 80-0063) and CuOx could be formed, as has been previously reported [45]. Accordingly, the impurity peak intensity in the composites increased as the amount of copper content increased. However, the characteristic diffraction peak of LaFeO3 at 32.3o weakened as the copper content increased in LCFO-5.’ We really appreciate your suggestions on our manuscript to help us improve the quality. We cherish your approval on our revised manuscript. We appreciate for reviewer’s warm advice earnestly. Special thanks to you for your good comments to help us improve the manuscript.
This manuscript is a resubmission of an earlier submission. The following is a list of the peer review reports and author responses from that submission.
Round 1
Reviewer 1 Report
I recommend seeing the comments in the PDF file.
Reviewer 2 Report
(1) Lines 16-17: pls provide the specific results of the effects of catalyst dosage…..
(2) Line 18: pls point out the kind of radical, sulfate radical or hydroxyl radical or both?
(3) Keywords: pls add LaCuxFe1-xO3 in front of perovskite.
(4) Some relevant references, such as 10.1016/j.cej.2020.128176, 10.1016/j.cclet.2021.10.087, 10.1016/j.seppur.2022.120716, 10.1016/j.cej.2022.138588 should be cited in introduction.
(5) Pls reduce the number of Figures for Physicochemical characteristics.
(6) Pls add more signals in figures 1-3 instead of the main text.
(7) Figure 4: What is the specific concentration of Cu and Fe ions? Maybe LCFO-4 other than catalyst is better in legend.
(8) Section 2.5: the reason for the degradation efficiency at pH 11 was further enhanced should be reconsidered, pls provide the zeta point of LCFO-4 and the leached metal ions at different pH and refer 10.1016/j.chemosphere.2016.02.089.
(9) Pls provide pH changes after HCO3- addition.
(10) What about the role of 1O2 (Eqs. 19 and 20), pls add some explanation.
(11) Line 259: the removal data of BPA from the LL and SW are different from figure 9b, pls revise.
Reviewer 3 Report
The authors discuss the catalytic activity of mixed Cu/Fe lanthanum perovskites. Overall, the study is of interest, however I have some serious concerns regarding first the synthesized perovskites, and second the missing catalytic tests on all but one perovskite.
In line 73-74 the authors state "The perovskite LaAxB1-xO3 (A or B= Co, Fe, Cu) is expected to be a promising catalytic material as a in SR-AOPs". However, several studies showed that LaFeO3 exhibits only limited activity. See e.g., https://doi.org/10.3390/catal12020187 and http://dx.doi.org/10.1016/j.ces.2016.11.017. Therefore, the authors need to justify their statement and the importance of this study.
In addition, the authors themselves state the perovskite to have the ABO3 structure, but claim the La2CuO4 structure to be a perovskite as well. Please describe why as the A2BO4 structure usually is considered a spinel phase.
I suggest the authors to perform Rietveld refinements on their X-ray diffractograms to gain more insight to the purity and crystal of their synthesized perovskites. The perovskites do not seem to be single-phase, and therefore, it will be difficult to explain if it is the perovskite or some other phases that migth impact the catalytic activity.
In Figure 2, please add the SEM image used to perform the EDX mapping.
Why is only LCFO-4 tested as catalyst? Why not compare all the synthesized perovskites - despite my concern about the purity?
The anion effect (Chloride and bicarbonate) needs more explanation and possible comparison to literature. Are these trends found elsewhere? Why is a low amount of Cl- promising? And why is the no trend for the HCO3-?
